# Cell-free assays reveal that the HIV-1 capsid protects reverse transcripts from cGAS immune sensing

Tiana M. Scott[1☯], Lydia M. Arnold[1☯], Jordan A. Powers[1], Delaney A. McCann[1], Ana B. Rowe[1], Devin E. Christensen[2], Miguel J. Pereira[2], Wen Zhou[3], Rachel M. Torrez[2], Janet H. Iwasa[2], Philip J. Kranzusch[4,5], Wesley I. Sundquist[2], Jarrod S. Johnson[1]*

1 Division of Microbiology and Immunology, Department of Pathology, University of Utah School of Medicine, Salt Lake City, Utah, United States of America, 2 Department of Biochemistry, University of Utah School of Medicine, Salt Lake City, Utah, United States of America, 3 Department of Immunology and Microbiology, School of Life Sciences, Southern University of Science and Technology, Shenzhen, China, 4 Department of Microbiology, Harvard Medical School, Boston, Massachusetts, United States of America, 5 Department of Cancer Immunology and Virology, Dana-Farber Cancer Institute, Boston, Massachusetts, United States of America

☯ These authors contributed equally to this work.
* jarrod.johnson@path.utah.edu

## Abstract

Retroviruses can be detected by the innate immune sensor cyclic GMP-AMP synthase (cGAS), which recognizes reverse-transcribed DNA and activates an antiviral response. However, the extent to which HIV-1 shields its genome from cGAS recognition remains unclear. To study this process in mechanistic detail, we reconstituted reverse transcription, genome release, and innate immune sensing of HIV-1 in a cell-free system. We found that wild-type HIV-1 capsids protect viral genomes from cGAS even after completing reverse transcription. Viral DNA could be "deprotected" by thermal stress, capsid mutations, or reduced concentrations of inositol hexakisphosphate (IP6) that destabilize the capsid. Strikingly, the capsid inhibitor lenacapavir also disrupted viral cores and dramatically potentiated cGAS activity, both in vitro and in cellular infections. Our results provide biochemical evidence that the HIV-1 capsid lattice conceals the genome from cGAS and that chemical or physical disruption of the viral core can expose HIV-1 DNA and activate innate immune signaling.

## Author summary

Human Immunodeficiency Virus type 1 (HIV-1) remains a global health threat. Drug therapies can help manage HIV-1 infection and prevent its spread, but without treatment, the immune system fails to control the virus. In part, this is because HIV-1 is thought to avoid detection by immune cells during entry. Once inside a target cell, the virus copies its RNA genome into double-stranded DNA within a viral protein shell called the capsid. In this study, we developed a novel cell-free method to study how the HIV-1 capsid shields viral DNA from detection. We found that the capsid normally functions as a molecular cloak that hides the viral DNA from innate immune sensors. We also

**Data availability statement:** The data that support the findings of this study are publicly available from Figshare with the identifier https://doi.org/10.6084/m9.figshare.c.7528029.v1.

**Funding:** We acknowledge funding from National Institutes of Health grant U54 AI170856 (to WIS and JSJ), R56 AI174930 (JSJ), R01 AI181713 (JSJ), 1DP2GM146250-01 (PJK), and grants from the Pew Biomedical Scholars program (PJK), and the Burroughs Wellcome Fund PATH program (PJK). W.Z. was supported in part through a Charles A. King Trust Postdoctoral Fellowship. The funders had no role in study design, data collection, and interpretation, or the decision to submit the work for publication.

**Competing interests:** The authors have declared that no competing interests exist.

identified experimental conditions and drug treatments that can "break" the capsid and allow the viral DNA to be detected, both in vitro and in cells. We anticipate that future work using cell-free systems will reveal how host factors regulate innate detection of HIV-1, possibly guiding new therapeutic strategies that block infection and engage antiviral immune responses simultaneously.

## Introduction

HIV-1 encapsidates the viral RNA genome and enzymes necessary for replication within a conical capsid comprising ~1,200 copies of the CA protein. Following viral fusion with a target cell membrane, the capsid and its contents (collectively known as the "core") are released into the cytosol. This initiates reverse transcription, which converts two copies of the single-stranded RNA genome into one copy of double-stranded DNA (dsDNA). Reverse transcription proceeds inside an intact, or largely intact, capsid as the viral core is trafficked from the cell surface to an integration site in the nucleus [1–10]. During a successful infection, reverse transcription likely completes in the nucleus [4,11–14], with capsid uncoating and genome release occurring only moments before viral integration into host chromatin [4,5]. Thus, the HIV-1 capsid is thought to shield the viral genome from cytosolic innate immune sensors during entry [15–17]. Yet, under some conditions, retroviral capsids fail to protect reverse transcribed viral DNA completely, permitting its detection in dendritic cells and macrophages and enabling the downstream induction of interferon (IFN) and inflammatory cytokines [16–21]. Mechanistic studies of HIV-1 sensing have remained challenging, however, because detection of reverse transcripts is a rare event that happens deep within these specialized cells.

Cyclic GMP-AMP synthase (cGAS) is the primary innate immune sensor that detects retroviral DNA [17,18]. Upon binding dsDNA in a sequence-independent manner, cGAS synthesizes the dinucleotide second messenger 2′3′-cyclic GMP-AMP (cGAMP), which binds and activates the adapter protein Stimulator of IFN genes (STING). STING then promotes downstream antiviral activities, including activation of the transcription factors IRF3 and NF-κB, which translocate to the nucleus and drive expression of type I and type III IFN [22,23]. IFN is rapidly secreted and initiates autocrine or paracrine signaling through IFN receptors. This upregulates hundreds of IFN-stimulated genes (ISGs), many of which can potently restrict HIV-1 infection [24–27]. Capsid structural integrity apparently influences the degree to which cGAS recognizes viral DNA, as specific CA mutations and CA-binding proteins are linked with increased or decreased IFN responses during infection [15–17,21,28–33]. However, the parameters that regulate exposure of viral DNA and its detection by cGAS remain unclear.

Here, we have established a cell-free system for studying the essential steps in cGAS-mediated sensing of HIV-1. We find that wild-type (wt) HIV-1 capsids remain stable and protect their genomes from cGAS detection, even several days after reverse transcription. We also identified conditions and drug treatments that "deprotected" viral genomes and increased cGAS activity in vitro and in cells. Thus, the HIV-1 capsid is a molecular cloak that shields the viral genome from innate immune sensing.

## Results

### Reconstitution of HIV-1 innate immune sensing in vitro

The discovery that HIV-1 viral capsids are stabilized by IP6 has facilitated biochemical studies of endogenous reverse transcription and host-virus interactions [3,34–36]. We sought to

extend these methods to study the physical and biochemical principles that control cGAS-mediated detection of reverse transcripts. To perform efficient endogenous reverse transcription (ERT), we permeabilized the viral membrane using the pore-forming peptide melittin, stabilized viral capsids in buffers that contained IP6 and ribonucleotides at the physiological levels observed in macrophages, and provided dNTP substrates for reverse transcription (Fig 1A). Samples were then incubated with recombinant human cGAS to determine how efficiently the ERT products activated cGAS to produce cGAMP. cGAS activation levels were compared to those produced by a range of naked plasmid DNA concentrations. Relative DNA levels were measured by quantitative PCR (qPCR) detection of plasmid and full-length HIV-$1_{SG3}$ genomes with primer sets that detected early, intermediate, and late reverse transcripts, and the equivalent DNA segments in the plasmid control (Fig 1B). ERT was highly efficient, as previously reported [3], yielding roughly 0.5 copies of late reverse transcripts per input viral core. Levels of early and intermediate products are higher because there are two early and intermediate primer binding sites on each copied genome, and because RT can initiate on both packaged RNA strands (before ultimately resolving to a single copy of dsDNA). As expected, plasmid DNA strongly activated cGAS in a concentration-dependent fashion. However, the HIV-1 ERT samples did not activate cGAS, suggesting that the viral DNA remained protected inside the capsid (Fig 1C).

Performing ERT and cell-free sensing assays in two stages allowed us to uncouple parameters that influence ERT from parameters that impact cGAS activity (Fig 1A). Viral cores that successfully completed RT were destabilized by limited heat treatment, which exposed the viral genome to cGAS (Fig 1D). Under these conditions, the viral DNA was sensed efficiently. In contrast, heat treatment of plasmid control samples did not alter cGAS activity (S1A–C Fig). Linear and circular forms of control DNA samples were detected at similar levels (S1D–F Fig), which agrees with previous findings using recombinant cGAS assays [37], and indicates that linear viral DNA and control circular plasmid DNA should be sensed equivalently if the viral DNA were fully exposed. Together, these data reinforce the idea that viral DNA is protected by the capsid.

HIV capsid stability can also be modulated by IP6 [34,35] or by mutations in CA [3,38–42]. We therefore tested the effects of both variables on cGAS sensing of ERT products. Reducing IP6 from the cellular 40 µM concentration [43] to 0.4 µM following reverse transcription increased cGAS detection of viral DNA by ~10-fold (Fig 1E). cGAS sensing was also altered by CA mutations that either stabilize (E45A) or destabilize (Q63/67A) capsids [3,38]. Specifically, we observed that the destabilizing Q63/67A mutant enhanced cGAS activation (normalized for cDNA levels), particularly as IP6 levels were reduced (Fig 1F and G). Conversely, the stabilizing E45A CA mutation reduced cGAS activation, resulting in limited cGAMP production even at very low IP6 levels. Both stabilized and destabilized capsid mutants were less infectious than wt HIV-1 when pseudotyped reporter viruses were compared in THP-1 monocytic cells (S1G Fig). Our in vitro results indicate that capsid stability modulates cGAS recognition, with stable capsids protecting viral DNA from cGAS recognition better than unstable capsids.

## Capsid inhibitors trigger viral core disruption and cGAS activation in vitro

Given that capsid stability regulates innate immune detection of HIV-1, we hypothesized that we could exploit chemical disruption of the capsid to unmask cGAS sensing of reverse transcripts. Consistent with this idea, the small molecule capsid inhibitor, PF74, can increase innate immune sensing of HIV-1 in cell lines, although this inhibitor has relatively low affinity for the capsid and only blocks viral replication at micromolar concentrations [15,44]. Advances in drug development have led to lenacapavir/SUNLENCA (LEN), an

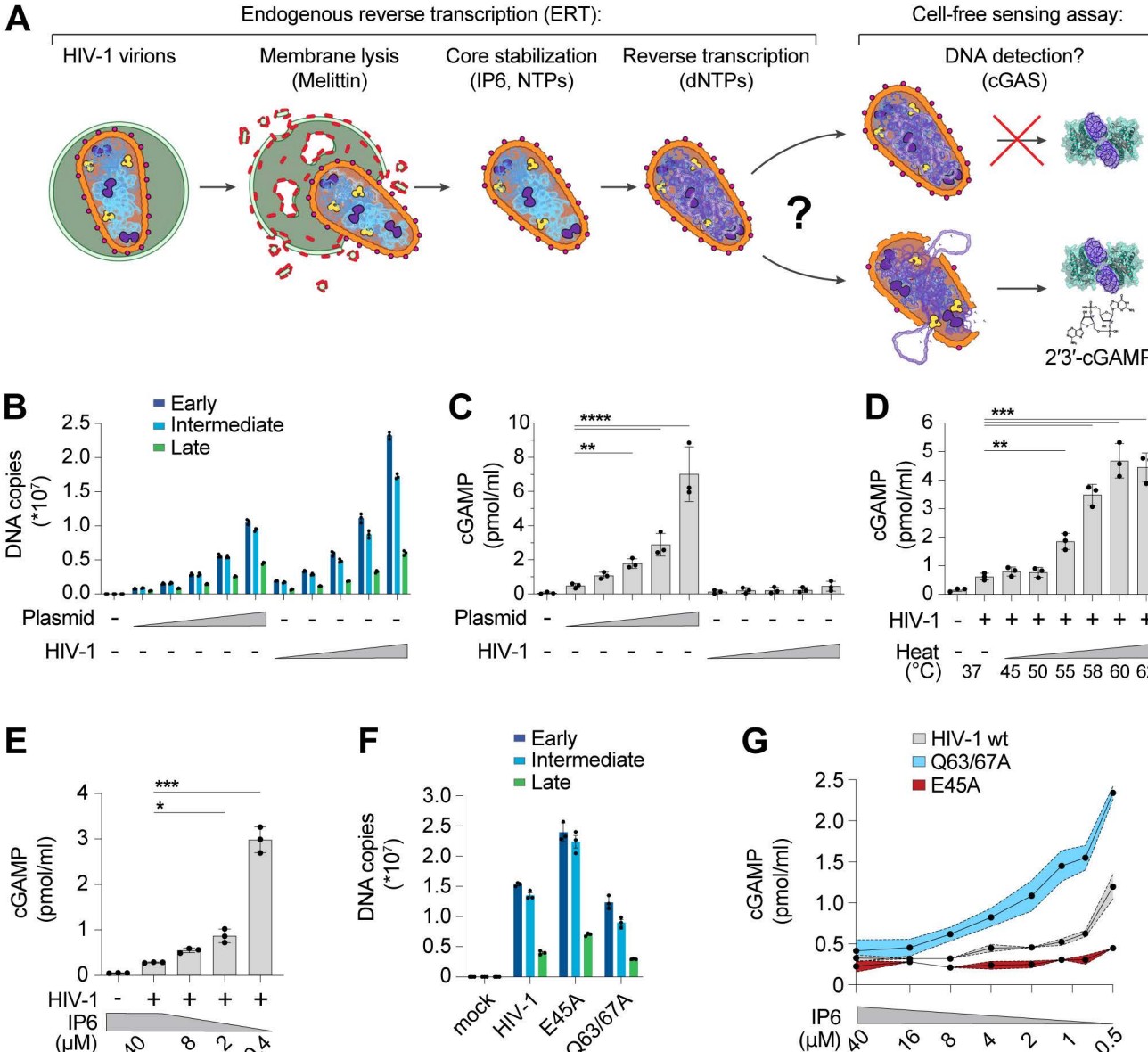

**Fig 1. The HIV-1 capsid protects RT products from cGAS in vitro. (A)** Steps in the endogenous reverse transcription (ERT) and cell-free sensing assays. Capsids that remain intact after ERT may protect viral DNA from detection and capsids that lose integrity may permit cGAS activation and cGAMP production. **(B)** qPCR measurements of plasmid DNA control containing the viral genome sequence compared to HIV-1 ERT reactions in 2-fold dilution series, incubated for 16 h under identical "standard" conditions at 37°C (including melittin, IP6, NTPs, dNTPs, MgCl$_2$, NaCl, and BSA in a Tris buffer to approximate cytoplasmic conditions as stated in the Methods). Primer sets detect sequences for early (minus strand strong stop), intermediate (first strand transfer), and late RT (second strand transfer), which are present in both the plasmid and viral genome. **(C)** Cell-free sensing assay for samples shown in (B). After ERT, samples were incubated with recombinant cGAS for 8 h at 37°C. cGAMP levels were determined by ELISA (this applies to all cell-free sensing assays). **(D)** Cell-free sensing assay of heat treated ERT samples. Samples from standard ERT reactions (16 h at 37°C) were held at the indicated temperatures (20 s) and then subject to cell-free sensing assays performed as described in the Methods section. **(E)** Samples from standard ERT reactions (16 h at 37°C) performed with cellular levels of IP6 (40 μM) were diluted at equivalent amounts into cell-free sensing assay reactions that contained a range of IP6 (40 μM to 0.4 μM final concentration). **(F)** qPCR measurements of DNA copy numbers from ERT reactions with wt HIV-1 virions compared to virions with stabilizing (E45A) or destabilizing (Q63/67A) mutations in CA. **(G)** Cell-free sensing assay of samples from (F) after normalizing for DNA input and diluting into cGAS reactions at the indicated IP6 concentrations. Statistics were calculated using a one-way ANOVA with Tukey's multiple comparisons test: p < 0.05: *, p < 0.01: **, p < 0.001: ***, p < 0.0001: ****. Graphs depict mean ± SD from three samples from a representative experiment, selected from three independent experiments.

FDA-approved capsid inhibitor that is 1,000-fold more potent than PF74 [45,46]. LEN binding reduces core particle elasticity [47] and stabilizes hexamer contacts [48,49], thereby flattening and rigidifying patches of the conical CA lattice and inducing ruptures [3,50]. We therefore tested whether LEN could "expose" viral reverse transcripts for detection in our vitro system.

Time course experiments to track production of reverse transcripts relative to naked plasmid DNA again showed robust recombinant cGAS activity for the plasmid control, but not ERT samples, despite comparable levels of plasmid and ERT DNA (Fig 2A). However, addition of LEN dramatically increased cGAS activation in the virus samples at late time points when dsDNA accumulated (Fig 2A, lower panel). After 16 h of ERT, LEN increased cGAS detection of HIV-1 DNA by more than an order of magnitude, and this effect was specific for the ERT products because LEN did not alter cGAS sensing of naked plasmid DNA.

To determine whether cGAS activation by HIV-1 samples incubated with LEN depended on reverse transcription, we performed ERT and cGAS reactions with increasing concentrations of the reverse transcriptase inhibitor efavirenz (EFV). As expected, EFV blocked RT product formation in a dose-dependent manner (Fig 2B, top panel). EFV treatments also reduced LEN-dependent cGAMP production, implying that cGAS activation reflected viral DNA sensing (Fig 2B, bottom panel). Similar effects were observed for two other treatments that blocked ERT; omitting dNTPs or introducing a point mutation that inactivated reverse

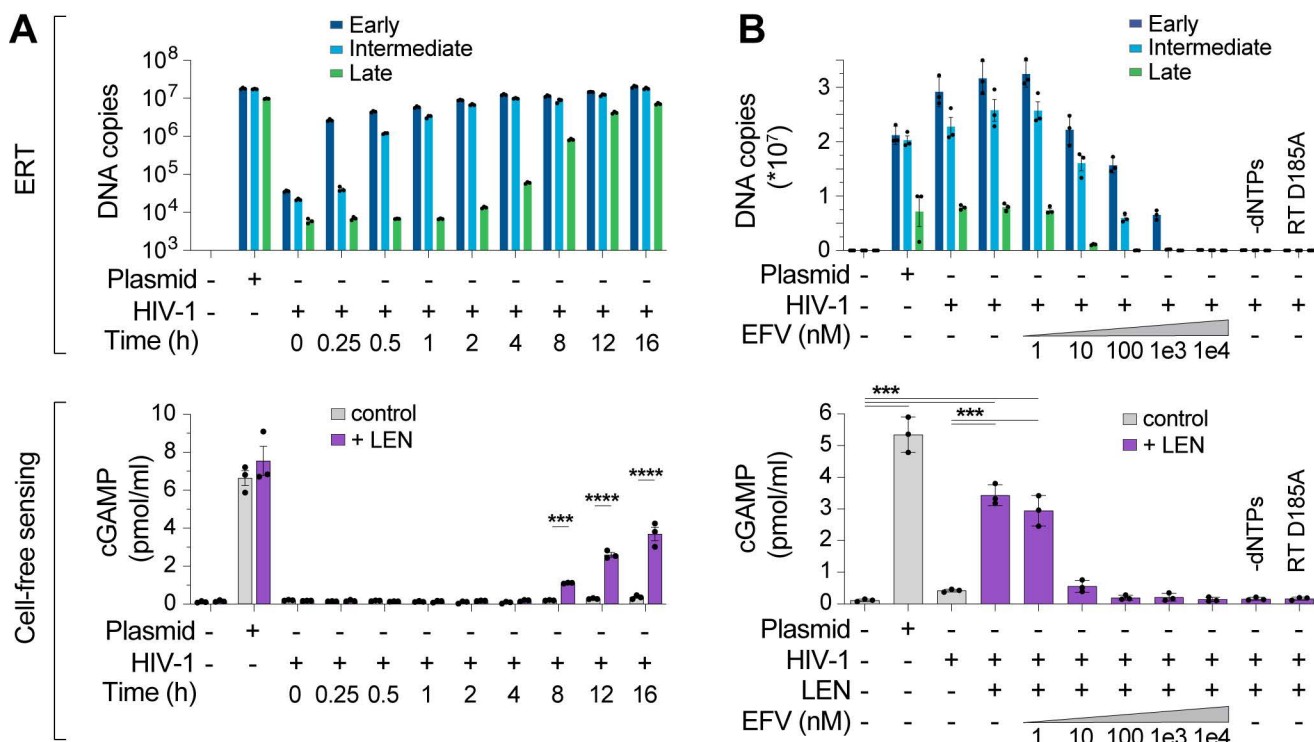

**Fig 2. Lenacapavir triggers cGAS sensing of HIV-1 reverse transcripts. (A)** DNA product levels from an HIV-1 ERT time course compared to a plasmid control, as measured by qPCR (top panel). After ERT, samples were incubated with recombinant cGAS ± LEN (100 nM) and cGAMP levels were determined by ELISA (cell-free sensing assays, bottom panel). **(B)** DNA products (top panel) from standard ERT reactions (16 h at 37°C) performed in the presence of increasing concentrations of the RT inhibitor efavirenz (EFV), or without dNTPs, or using a virus containing an inactivating mutation in RT (D185A). After ERT, samples were reacted with recombinant cGAS ± LEN (100 nM) and cGAMP levels were determined by ELISA (bottom panel). Statistics were calculated using a one-way ANOVA with Tukey's multiple comparisons test: p < 0.001: ***, p < 0.0001: ****. Graphs depict mean ± SD from three samples from a representative experiment, selected from three independent experiments.

transcriptase (D185A) (Fig 2B). Dropout experiments confirmed that, as expected, LEN-dependent cGAS sensing of HIV-1 RT products also required virion permeabilization (melittin), ERT (MgCl$_2$ and IP6), and enzymatic cGAMP production (NTPs and cGAS) (S2A–E Fig). We note that higher concentrations of MgCl$_2$ were associated with greater cGAS activity at baseline and reduced LEN responsiveness. We speculate that these MgCl$_2$ effects could be driven by two factors: 1) cations such as Mg$^{2+}$ and Mn$^{2+}$ support cGAS activity in vitro [51], and 2) at high concentrations, Mg$^{2+}$ cations might disrupt capsid stability, by interfering with CA subunit interactions and/or by chelating IP6 away from the capsid, leading to loss of capsid integrity, genome release, and reduced LEN responsiveness. Importantly, cGAS sensing was also triggered by the LEN analog, GS-CA1 [45], and by the weaker PF74 inhibitor [52], which all target the same capsid binding pocket [46,53] (S2F and G Fig). Together, these data indicate that: 1) the HIV-1 capsid protects the viral genome from innate immune sensing, and 2) pharmacological disruption of the capsid enables cGAS detection of reverse transcripts.

We next assessed the specificity of the LEN effects by testing activity against CA mutations that inhibit LEN binding and confer drug resistance [46,54]. Control experiments confirmed that the M66I and Q67H/N74D CA mutations conferred LEN resistance to reporter virus infection of THP-1 cells, as expected (S3A Fig). In vitro, HIV-1 cores carrying these LEN resistance mutations completed ERT at normal levels (Fig 3A). However, the mutant virions did not stimulate cGAS activity when treated with LEN (Fig 3B). In control experiments, heat treated mutant capsids stimulated cGAS normally, indicating that their DNA was detectable when exposed by other methods of capsid disruption (S3B Fig). These experiments demonstrate that LEN binds the viral capsid to stimulate cGAS activity.

During a normal infection, the viral DNA must eventually exit the capsid to integrate into the host genome, but the mechanism of release (termed "uncoating") is not well understood. In particular, it is not yet clear whether uncoating occurs stochastically, is driven by pressure caused by accumulation of viral dsDNA [55], or is facilitated by host factors. To determine whether prolonged incubation of viral cores might lead to capsid permeability and genome release, we ran ERT reactions and incubated the samples for 16, 24, 36, 48, and 72 h at 37°C and then performed cGAS DNA sensing assays. DNA product levels were quite stable across these time points (Fig 3C), and cGAS activation increased modestly (3- to 4-fold) from 16 to 72 h, indicating that ERT products became more exposed over time (grey bars, Fig 3D). Nevertheless, even after 72 h, LEN treatment further stimulated cGAS activity (~3-fold), indicating that capsids provide stable protection of viral DNA in vitro for days after completion of reverse transcription.

## LEN treatment triggers cellular sensing of HIV-1 via the cGAS-STING pathway

We also extended our findings to cellular models of HIV infection. In vitro, LEN-dependent cGAS activation by ERT reactions exhibited biphasic behavior (Fig 4A). At high concentrations (>30 nM), LEN inhibited reverse transcription (Fig 4A, top panel) and cGAS activation was therefore prevented because dsDNA was not present for detection [3]. Lower LEN concentrations (1–10 nM) stimulated cGAS activation in a dose dependent fashion, consistent with the idea that reverse transcription proceeded, and viral cores were increasingly disrupted, triggering cGAS activation (Fig 4A, bottom panel). To determine whether these effects were recapitulated in the context of a viral infection, we tested the effects of different LEN levels on THP-1 cells treated with HIV-1-GFP and tracked both infectivity and innate immune activation, as measured by induction of two different interferon-stimulated genes (ISGs): ISG15 and SIGLEC1 (Fig 4B and C). We found that high LEN levels inhibited cGAS activation (likely due

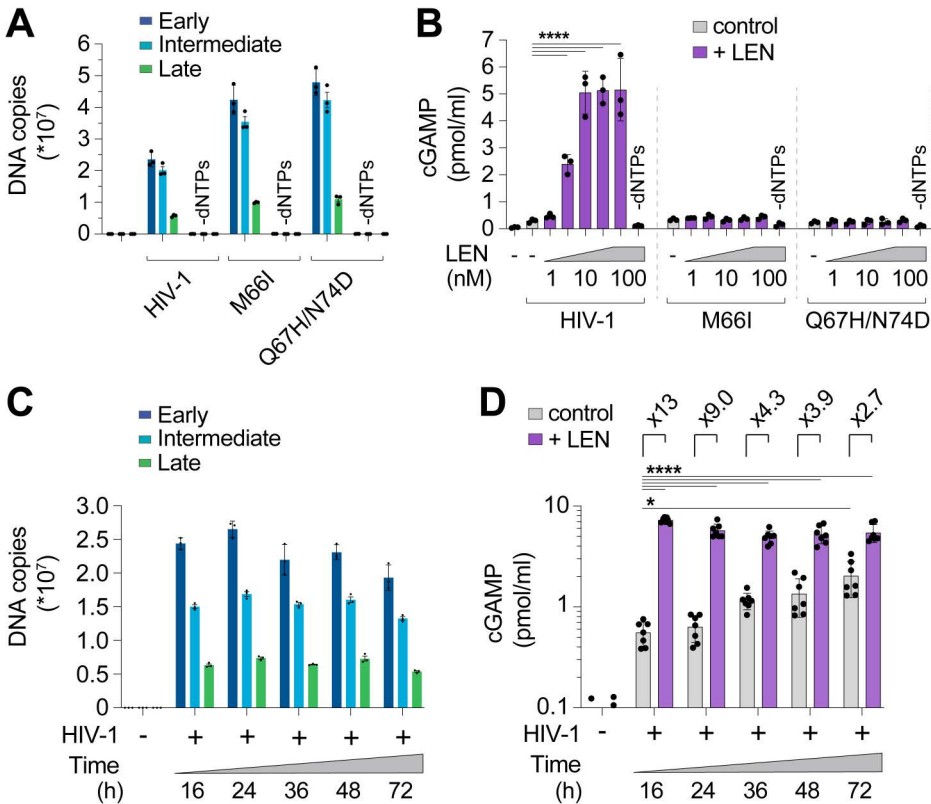

**Fig 3. CA mutations confer resistance to LEN-triggered cGAS activation. (A)** qPCR measurements of DNA copy number from ERT reactions under standard conditions (16 h at 37°C) with wt HIV-1 virions compared to virions bearing LEN resistance-associated mutations in CA (M66I and Q67H/N74D), performed with or without dNTPs. **(B)** Cell-free sensing assay of wt, M66I, or Q67H/N74D samples from (A). After ERT, samples were incubated with recombinant cGAS and increasing amounts of LEN (0, 1, 3.2, 10, 32, 100 nM), or LEN (100 nM) −dNTPs. cGAMP levels were determined by ELISA. **(C)** qPCR measurements of DNA copy number for ERT reactions that were held at 37°C for the indicated times. **(D)** Cell-free sensing assay of ERT samples from (C). After ERT, samples were normalized for late RT DNA copy number and then incubated with recombinant cGAS under standard conditions (± LEN at 100 nM). Brackets above graph represent the fold increase in cGAS activity with LEN at each time point. Statistics were calculated using a one-way ANOVA with Tukey's multiple comparisons test: $p < 0.05$: *, $p < 0.0001$: ****. Graphs depict mean ± SD from three measurements from a representative experiment, selected from 3 independent experiments (A–C) or from 7 independent samples measured on different days (D).

to reduced reverse transcription, as described below), but lower LEN levels stimulated ISG15 and SIGLEC1 expression (Figs 4C and S4A–E), at concentrations that matched the levels that stimulated cGAS activity in vitro (1–10 nM). Thus, the LEN effects observed in cells mirror those observed in vitro.

We confirmed that high concentrations of LEN (>30 nM) reduced early, intermediate, and late HIV-1-GFP reverse transcripts in THP-1 cells (Fig 5A), which agrees with recently published work [33]. Our dose response matrix of virus and LEN concentrations indicated that potent innate immune responses could be revealed with low virus input and therapeutic levels of LEN (e.g., 10 nM, S4A and B Fig). To validate these findings further and evaluate the kinetics of the innate response, we tested HIV-1-GFP at low mutliplicity of infection (MOI) and measured ISG expression at different times after infection. THP-1 cells began to express GFP roughly 24 h after exposure to low levels of HIV-1-GFP (12.5 nM p24; MOI < ~0.5) but did not express high levels of ISGs during the two-day time period tested (Fig 5B). In contrast, the

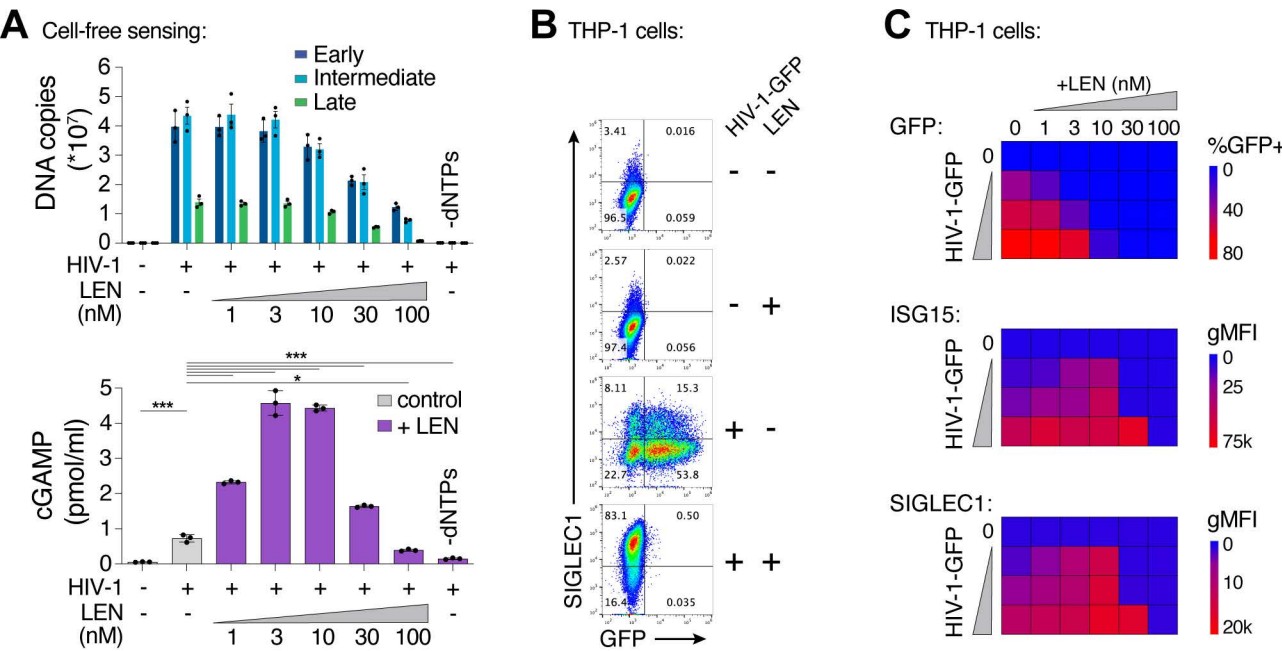

**Fig 4. LEN potentiates sensing of HIV-1 in vitro and in cells.** (A) qPCR measurements of DNA copy number from ERT reactions performed under standard conditions (16 h at 37°C) with or without LEN at the indicated concentrations (top panel). After ERT, samples were reacted with recombinant cGAS in cell-free sensing assays (bottom panel), maintaining the same concentrations of LEN as used in ERT. (B) Flow cytometry of THP-1 monocytic cells infected with HIV-1-GFP (12.5 nM p24) for 48 h (± LEN at 10 nM), showing expression of GFP (viral infectivity) and SIGLEC1 (IFN response). (C) Heat maps of GFP, ISG15, and SIGLEC1 expression as determined by flow cytometry of THP-1 cells infected with HIV-1-GFP for 48 h. Columns show a range of LEN concentrations and rows show increasing virus doses (12.5, 25, and 50 nM p24). n = 3. For (A), statistics were calculated using a one-way ANOVA with Tukey's multiple comparisons test: $p < 0.05$: *, $p < 0.001$: ***, $p < 0.0001$: ****. Graphs depict mean ± SD from three samples from a representative experiment, selected from 2–3 independent experiments, unless otherwise noted.

same virus dose in the presence of 10 nM LEN triggered strong induction of a series of ISGs (*IFITM1, IFITM3, MX1, ISG15*, and *IFI27*), indicating that LEN exposed viral DNA above the threshold required for cGAS-STING activation.

LEN treatments did not stimulate ISG expression upon infection by the LEN-resistant mutants M66I and Q67H/N74D, showing the specificity of drug activity in cells (S4F Fig). We note, however, that although the M66I mutation reduced viral infectivity, the mutant virus induced ISG expression at levels that were similar to wt HIV-1, consistent with the idea that this mutation reduces viral fitness owing to capsid structural defects [49,56]. Most importantly, our data demonstrate that LEN can stimulate innate immune sensing of incoming wt HIV-1 particles, at therapeutically relevant concentrations that still block infection.

To determine whether LEN-triggered innate immune activation occurred through the canonical cGAS/STING pathway, we employed the lentiCRISPR system [57] to knock out the key pathway components, cGAS, STING, or IRF3. Pools of THP-1 monocytic cells were transduced with lentiCRISPR vectors, expanded for two weeks under antibiotic selection, and target protein loss was confirmed by immunoblot (Fig 6A). As expected, HIV-1 infection was blocked by LEN treatment (10 nM), but was not affected by loss of cGAS, STING, or IRF3 (Fig 6B). However, loss of each pathway component substantially reduced ISG protein expression, both in the absence and presence of LEN (Fig 6C). Thus, LEN stimulates innate immune recognition of HIV-1 through the canonical cGAS/STING pathway.

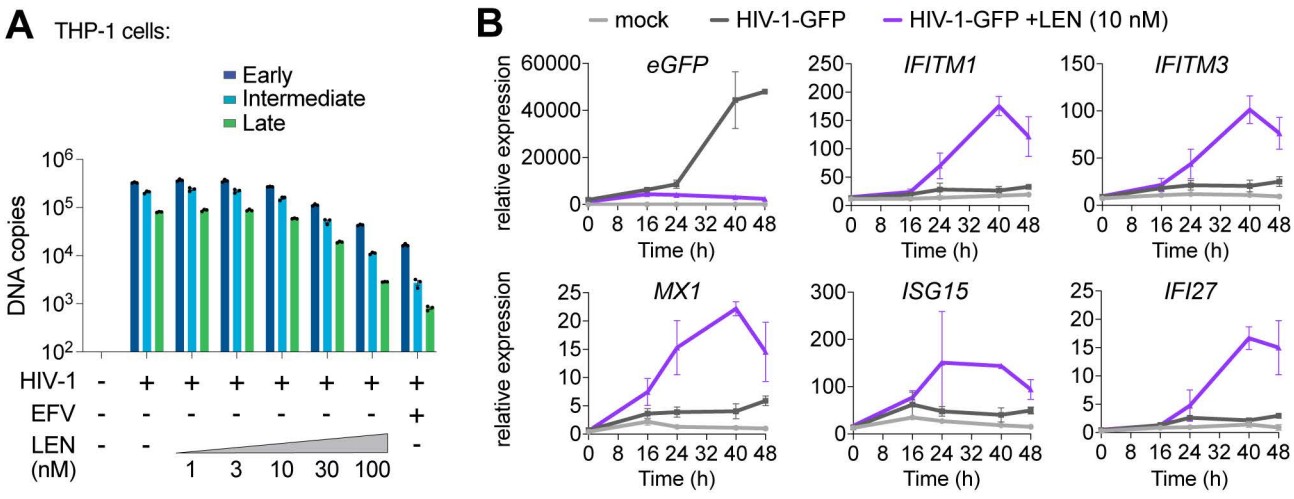

**Fig 5. Therapeutic concentrations of LEN activate innate immune sensing at low MOI. (A)** qPCR measurements of reverse transcription in THP-1 monocytic cells that were infected for 24 h with HIV-1-GFP (12.5 nM p24, corresponding to an MOI of < 0.5) with increasing concentrations of LEN. EFV (10 μM) was included as a positive control to inhibit RT. **(B)** Time course of gene expression in THP-1 monocytic cells infected with HIV-1-GFP (12.5 nM p24) with or without LEN (10 nM). Graphs show qPCR measurements of relative expression for *eGFP, IFITM1, IFITM3, MX1, ISG15*, and *IFI27* at 0, 16, 24, 40, and 48 h time points. Graphs depict mean ± SD from three samples from a representative experiment, selected from 3 independent experiments.

Recent work has suggested that an additional factor, polyglutamine-binding protein 1 (PQBP1), helps to license cGAS activity during HIV-1 infection, through its capacity to link cGAS to the viral capsid [32,58]. In our cell-free system, full-length recombinant PQBP1 protein did not enhance cGAS activity, either alone or together with HIV-1 (S5A and B Fig). Similarly, we observed that loss of PQBP1 did not alter viral infectivity or reduce innate immune sensing of HIV-1 in myeloid cells, using two independent guide RNAs to knockout PQBP1 expression (S5C–G Fig). Loss of PQBP1 also had no effect when innate immune sensing was limited to early steps of the life cycle in the presence of the integration inhibitor elvitegravir (S5F and G Fig). In contrast, cGAS removal (positive control) ablated ISG responses to HIV-1-GFP under all conditions tested.

Finally, we evaluated whether capsid inhibition could facilitate innate immune responses while blocking HIV-1 infection in primary human myeloid cells. Human monocyte-derived dendritic cells (MDDCs) and monocyte-derived macrophages (MDMs) can be infected by HIV-1, particularly when the lentiviral accessory protein Vpx is provided in trans to overcome SAMHD1-mediated restriction in MDDCs [59,60]. In some cases, MDDCs and MDMs can respond to HIV-1 by producing type I IFN and other inflammatory cytokines [16,17,19,21,61,62]. We infected MDDCs and MDMs from four unique donors with HIV-1-GFP (Fig 6D and F), and found that 3–30 nM concentrations of the LEN analog, GS-CA1, efficiently blocked infection in MDDCs and MDMs. Importantly, however, ISG15 activation was still sustained at these inhibitory concentrations (Fig 6E and G). Hence, potent LEN-class capsid inhibitors can simultaneously prevent HIV-1 infection while still maintaining robust innate immune responses in primary human myeloid cells.

## Discussion

IFN creates strong selective pressure on HIV-1 during sexual transmission and during viral rebound in response to withdrawal of antiviral therapy [63,64]. It is therefore critical to

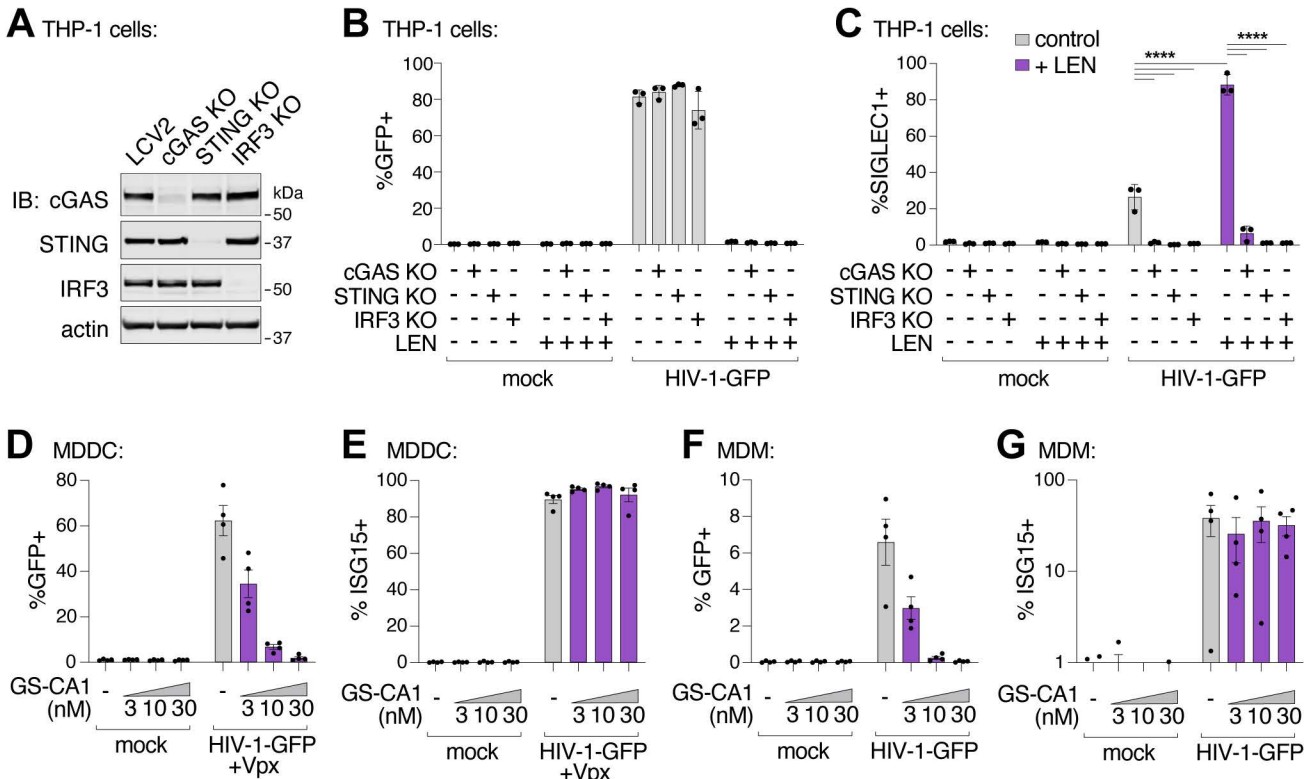

**Fig 6. Capsid inhibitors potentiate sensing of HIV-1 through the cGAS-STING pathway in cells. (A)** Immunoblots of THP-1 lysates after lentiCRISPR editing, depicting knockout of cGAS, STING, or IRF3 relative to a non-targeting vector control (LCV2). **(B and C)** Flow cytometry of lentiCRISPR-modified THP-1 cells infected with HIV-1-GFP (25 nM p24) for 48 h, showing the percent of cells positive for GFP (B) or SIGLEC1 (C) expression for the indicated conditions (± LEN at 10 nM). **(D and E)** Flow cytometry of MDDCs (derived from 4 separate donors) that were infected with HIV-1-GFP (50 nM p24) for 48 h with increasing concentrations of GS-CA1 (0, 3, 10, and 30 nM) added at the time of infection. Vpx was included to overcome SAMHD1-mediated restriction. Panels show infectivity (GFP+, D) and IFN induction (ISG15+, E). **(F and G)** Flow cytometry of MDMs (derived from 4 donors) that were infected with HIV-1-GFP (50 nM p24) in the same fashion as in (D–E) with increasing concentrations of GS-CA1 (0, 3, 10, and 30 nM) added at the time of infection. Infections were performed without Vpx. Graphs depict the percentage of GFP (D,F) or ISG15 (E,G) positive cells. For **(C)**, statistics were calculated using a one-way ANOVA with Tukey's multiple comparisons test: $p < 0.05$: *, $p < 0.001$: ***, $p < 0.0001$: ****. Graphs depict mean ± SD from three samples from a representative experiment, selected from 2–3 independent experiments, unless otherwise noted.

understand how HIV is sensed by the innate immune system. Multiple innate immune sensors can contribute to IFN and inflammation during HIV-1 infection [61,62,65–69], but cGAS activation is increasingly appreciated as a driver of protective anti-HIV immune responses. For example, this pathway is robustly activated in myeloid cells from elite controllers [70,71], and myeloid cell programming of polyfunctional CD8+ T cells is critical for control of infection [72,73]. However, precisely how viral DNA is released from the capsid to activate cGAS has not been well characterized.

To begin to address this issue, we have reconstituted endogenous reverse transcription, capsid destabilization, and innate immune sensing of HIV-1 in a cell-free system. We find that HIV-1 capsids protect reverse transcripts from detection by cGAS, and that this protection can last for days after reverse transcription is complete. This protection appears to be mediated by intact or largely intact capsids because cGAS recognition increases dramatically when capsids are disrupted by heat, reduced IP6 concentrations, or destabilizing mutations in the viral capsid. Our observation that capsid integrity and stability can strongly modulate

cGAS-mediated detection of HIV-1 is consistent with a growing body of literature indicating that the capsid sequence has a substantial impact on sensitivity to innate immune sensors [16,17,21,28,30], and that IP6 perturbations or improper capsid formation can promote cGAS activation [15,29]. We suggest that our in vitro system can be useful in characterizing these effects, and also in determining how host factors promote capsid uncoating and cDNA exposure.

Our work demonstrates that wt HIV-1 capsids have the intrinsic property of shielding viral DNA from innate immune detection, but we note that recruitment of host factors likely serves to further modify capsid stability in cells (reviewed in [74,75]). For example, cyclophilin A (CypA) [76] and CPSF6 [77] are well-characterized CA-binding proteins that play different roles during post-entry replication steps and may positively or negatively affect the capsid, as they can promote trafficking [78], influence nuclear entry [79,80], and affect integration site selection [81,82]. Another protein, TRIM5α from rhesus macaques can form a hexagonal cage surrounding the HIV-1 capsid [83] and block early steps in infection [84,85], potentially by disrupting core integrity or other mechanisms [86]. Moving forward, our cell-free assay will likely be useful for determining whether CypA, CPSF6, TRIM5α, and other capsid-binding factors have a direct effect on core stability. In this study, we tested the role of PQBP1, as it has been reported to bind to both the capsid and to cGAS, and to be important for innate immune sensing of HIV-1 [32,58]. Although we did not observe an effect with PQBP1 in vitro or validate that it is required for HIV-1 detection in myeloid cells, we cannot rule out that PQBP1 plays an important role under different circumstances.

Importantly, we found that capsid inhibitors of the LEN class can promote HIV-1 DNA genome exposure and cGAS activation in vitro, in THP-1 cells, and in primary human myeloid cells. In the THP-1 case, and presumably also in primary myeloid cells, cGAS activation by reverse transcription products leads to innate immune activation via the canonical STING pathway (Fig 6), which supports the growing body of literature evoking the importance of this pathway in HIV innate immunity [15–19,28–31,33,58,70,71]. However, we have not yet identified conditions that lead to LEN-induced IFN production in CD4+ T cells, suggesting that additional host circuitry might be required to unleash this response in the main target cells of HIV [87,88].

In summary, our experiments are consistent with a model in which viral cores enter the cytosol of myeloid cells, and reverse transcription normally proceeds inside intact or largely intact capsids, where viral cDNA is protected from innate immune sensors [1,16,21,31]. However, in the presence of LEN, accumulating reverse transcripts can become exposed by drug-induced capsid rupture, enabling viral DNA detection and downstream activation of IFN production via the cGAS-STING-IRF3 pathway (S1 Movie). Importantly, LEN can inhibit multiple steps in the virus life cycle at different drug concentrations, including virus assembly and a step corresponding to nuclear entry or a later pre-integration step, with the latter being the most potent block to infection in cultured cells (0.5–5 nM) [45]. Reverse transcription can also be inhibited, but this requires drug concentrations (≥25 nM) that are much higher than the protein-adjusted EC95 required to inhibit HIV-1 replication (~4 nM) [45,46]. We found that LEN promotes cGAS activation at concentrations that parallel those that block nuclear entry and/or integration (1–10 nM, see Fig 4), and we hypothesize that the most potent mechanism of drug action may be associated with premature genome exposure. Our experiments also indicate that, at least in culture, cGAS activation can occur under conditions where viral infectivity is suppressed but innate immune responses are maintained. We therefore speculate that the drug-dependent effects we observe in cultured cells might be harnessed to potentiate vaccination responses under conditions where viral replication is blocked while antiviral immunity remains engaged.

## Methods

### Ethics statement

This work meets the definition of research not involving human subjects as described in 45 CFR 46.102(f), 21 CFR 56.102(e) and 21 CFR 812.3(p) and satisfies the Privacy Rule as described in 45CFR164.514. Regarding the use of human cells and cell lines, no identifiable information is associated with the data obtained from our research.

### Cell lines and blood-derived dendritic cells

To generate MDDCs and MDMs, we acquired leukocytes from de-identified normal human donors from ARUP Blood Services, Sandy, UT, USA, similar to previously described methods [89]. We cannot report on the sex, gender, or age of the donors since the samples were de-identified and donors remain anonymous. Peripheral blood mononuclear cells (PBMCs) were layered over Ficoll-Paque Plus (GE Healthcare). CD14+ monocytes from PBMC buffy coats were isolated with anti-human CD14 magnetic beads (Miltenyi) and cultured in RPMI containing 10% heat-inactivated fetal bovine serum (FBS, Gibco), 50 U/mL penicillin, 50 µg/mL streptomycin (pen/strep, Thermo Fisher), 10 mM HEPES (Sigma), and 55 µM 2-Mercaptoethanol (Gibco). We tested multiple lots of FBS to identify batches that lead to minimal baseline induction of activation markers over the course of MDDC differentiation. MDDCs were derived in the presence of recombinant human GM-CSF (Peprotech) at 10 ng/mL and IL-4 at 50 ng/mL. MDMs were derived using recombinant human M-CSF (Peprotech) at 20 ng/mL. Fresh media and cytokines were added to cells (40% by volume) one day after CD14+ cell isolation. On day 4, MDDCs were collected, resuspended in fresh media with cytokines used for infection or stimulation such that experimental end points would fall on day 6. MDM cultures were replenished with fresh media on day 4, then infected or stimulated on day 5 so that experimental end points would fall on day 7 after isolation. MDDC and MDM experiments were performed using biological replicates from blood-derived cells from multiple individual donors as indicated in the figure legends.

We cultured 293FT cells (immortalized from a female fetus, Life Technologies Cat# R70007, RRID: CVCL_6911) in Dulbecco's modified Eagle's medium (DMEM, Thermo Fisher) supplemented with 10% heat-inactivated FBS, pen/strep, 10mM HEPES, and 0.1 mM MEM non-essential amino acids (Thermo Fisher). THP-1 cells (derived from a male patient with acute monocytic leukemia, ATCC) were cultured in RPMI (Thermo Fisher) containing 10% FBS, pen/strep, 10 mM HEPES, and 2-Mercaptoethanol.

Cell lines were used at early passage numbers and fresh stocks were thawed every 1–2 months. Cell cultures were routinely tested for mycoplasma contamination every 6 months. All cells were maintained at 37°C and 5% $CO_2$.

### Plasmids and mutagenesis

For endogenous reverse transcription reactions, HIV-1$_{SG3}$ was produced from pSG3Δenv plasmid, which was acquired from the NIH AIDS Reagent Program (Catalog # 11051) and contains mutations in the *env* gene rendering it non-functional (ΔEnv), but otherwise represents the full-length HIV-1 genome. HIV-1-GFP is derived from an HIV-1 NL4-3 vector that has been used to study immune responses in myeloid cells lines and primary human MDDCs [21]. pHIV-1-GFP is *env- vpu- vpr- vif- nef-*, with the GFP open reading frame in place of *nef*. We generated virus like particles packaging Vpx from the plasmid pSIV3+ (based on SIV-mac251, GenBank acc. no. M19499), which has been described previously [90]. These and all other plasmids used in this study are listed in S1 Table and have been made available through

Addgene. Target sequences for lentiCRISPR vectors are also provided (S1 Table). All lentiviral constructs were transformed into Stbl3 bacteria (Thermo Fisher) for propagation of plasmid DNA. The coding sequence for full-length human PQBP1 was synthesized as a GBlock gene fragment (IDT) and cloned into pCA528 by Gibson Assembly. All plasmids were prepped and purified through PureLink HiPure Plasmid Maxiprep Kits (Invitrogen).

## Virus and vector production

Reporter viruses, recombinant lentiviral vectors, and Vpx-containing virus-like particles, were produced by transient transfection in 293FT cells, similar to previously described methods [19]. Briefly, the day before transfection, cells were seeded onto poly-L-lysine- (Sigma) coated 15 cm plates to be ~70% confluent at the time of transfection. Cells were transfected with a total of 22–25 µg DNA using PEImax (Polysciences, Inc.). Each batch of PEI was titered to identify the optimal ratio of DNA:PEI for transfection (which was typically close to 1:2). To produce wt SG3ΔEnv and mutant virions, a single plasmid was transfected into 293FT cells. For HIV-1-GFP reporter viruses, we transfected 19.1 µg of the HIV cassette and 3.4 µg of pCMV-VSV-G. For lentiCRISPR vectors, we transfected 13.5 µg of the transgene plasmid, 8 µg of psPax2 helper plasmid, and 3.5 µg pCMV-VSV-G. Virus-like particles containing Vpx were produced by transfecting 19.1 µg pSIV3+ and 3.5 µg pCMV-VSV-G. The morning after transfection, cells were washed once and replenished with 30 mL of fresh media. Supernatants were harvested 36–48 h after replenishing media and then passed through 0.45 µm syringe filters (Corning) to remove debris.

To purify wt and mutant SG3ΔEnv virions used for ERT reactions and cell-free sensing assays, virus supernatants (30 mL) were treated for 1 h at 37°C with 2,500 Units ultrapure benzonase nuclease (Sigma, E8263-25KU) and subtilisin [91] (Sigma). Protease activity was then quenched with PMSF. Virus preparations were further purified through methods that will be described in a forthcoming publication. HIV-1-GFP wt and mutant virions were treated in a similar fashion except were not treated with subtilisin and PMSF. Supernatants were layered over a 20% sucrose cushion (in PBS) and concentrated by ultracentrifugation by spinning in 25 x 89 mm ultraclear tubes (Beckman) at 28 K rpm for 2 h at 4°C in an SW32 swing-bucket rotor (Beckman). SG3ΔEnv virions were thoroughly resuspended using 500 µL Hepes/Salt buffer per tube. HIV-1-GFP wt and mutant virions and lentiCRISPR vectors were resuspended in 1 mL of complete RPMI media. With high titer stocks of HIV-1-GFP and lentiCRISPR vectors we occasionally observed insoluble virus aggregates that we clarified from virion preparations by low-speed centrifugation at 300 rcf for 3 min at 4°C. Purified viral stocks were frozen at −80°C, titered by p24 ELISA (Express Bio), and used as indicated below.

## Bacterial expression and purification of recombinant human cGAS

Full-length human cGAS (aa 1–522, HGNC:21367) was produced and purified as previously described [92,93]. Briefly, human cGAS was expressed as a fusion protein with a 6×His-SUMO2 tag in *Escherichia coli* (*E. coli*) BL21-RIL DE3 (Agilent) bacteria harboring a pRARE2 tRNA plasmid. Initial cultures were grown in 30 mL of MDG media (25 mM $Na_2HPO_4$, 25 mM $KH_2PO_4$, 50 mM $NH_4Cl$, 5 mM $Na_2SO_4$, 2 mM $MgSO_4$, 0.5% glucose, 0.25% aspartic acid, 100 µg/mL ampicillin, 34 µg/mL chloramphenicol, and trace metals) to an $OD_{600}$ of ~0.05. These cultures were then used to seed 2 L M9ZB media (7.8 mM $Na_2HPO_4$, 22 mM $KH_2PO_4$, 18.7 mM $NH_4Cl$, 85.6 mM NaCl, 2 mM $MgSO_4$, 0.5% glycerol, 1% Casamino acids, 100 µg/mL ampicillin, 34 µg/mL chloramphenicol, and trace metals) at 37°C until an $OD_{600}$ of ~2.5–3.0 was reached. The cultures were supplemented with Isopropyl ß-D-1-thiogalactopyranoside

(IPTG) (0.5 mM) and incubated at 16°C for an additional 12–16 h. Next, bacteria were pelleted, flash-frozen in liquid nitrogen, and stored at −80°C until further purification.

Bacterial pellets with cGAS fusion proteins were resuspended in lysis buffer (20 mM HEPES-KOH pH 7.5, 400 mM NaCl, 10% glycerol, 30 mM imidazole, 1 mM dithiothreitol (DTT) and subsequently lysed by sonication. The supernatant containing cGAS was collected for initial purification using Ni-NTA (Qiagen) resin and gravity chromatography. Eluted 6×His-SUMO2-cGAS proteins were supplemented with ~250 μg of human SENP2 protease to remove the His-SUMO2 tag and treated in dialysis buffer (20 mM HEPES-KOH pH 7.5, 300 mM NaCl, 1 mM DTT) at 4°C for ~12 h. The dialysate was purified by using heparin HP ion exchange (GE Healthcare) and eluted with a 300–1,000 mM NaCl gradient. Full-length untagged cGAS protein was then ultimately purified using size-exclusion chromatography via a 16/600 Superdex S75 column (GE Healthcare). Purified cGAS was concentrated to ~10 mg/mL in storage buffer (20 mM HEPES-KOH pH 7.5, 250 mM KCl, 1 mM TCEP), flash-frozen in liquid nitrogen, and stored as aliquots at −80°C for biochemical experiments.

## Bacterial expression and purification of recombinant human PQBP1

Full-length human PQBP1 (HGNC:9330) was cloned into pCA528 for bacterial expression as a (His)$_6$-SUMO-fusion protein. (His)$_6$-SUMO-PQBP1 was expressed in BL21 RIPL cells grown in 2xYT media. Transformed cells were initially grown for 3–6 h at 37°C to an OD600 of 0.4–0.6, IPTG was added to final concentration of 0.5 mM to induce recombinant protein expression and cells were cultured for an additional 4 h. Cells were harvested by centrifugation at 6,000 x g and cell pellets were stored at −80°C.

All purification steps were carried out at 4°C except where noted. Frozen cell pellets were thawed and resuspended in lysis buffer: 50 mM Tris (pH 8.0 at 23 °C), 500 mM NaCl, 2 mM Imidazole, 0.5 mM EDTA, supplemented with lysozyme (25 μg/mL), PMSF (100 μM), Pepstatin (10 μM), Leupeptin (100 μM), Aprotinin (1 μM), and DNase I (10 μg/mL). Cells were lysed by sonication and lysates were clarified by centrifugation at 40,000 x g for 45 min. Clarified supernatant was filtered through a 0.45 μM PES syringe filter and incubated with 10 mL of cOmplete His-Tag purification beads (Roche) for 1 h with gentle rocking. Beads were washed with 150 mL unsupplemented lysis buffer. PQBP1 fusion protein was eluted with 50 mL of lysis buffer supplemented with 250 mM Imidazole.

Eluted proteins were treated with 100 μg His$_6$-ULP1 protease overnight in 6–8 k MWCO dialysis bags while dialyzing against 2 x 2 L of 25 mM Tris (pH 8.0 at 23°C), 50 mM NaCl, 1 mM TCEP. The dialysate was purified by Q Sepharose chromatography (HiTrap Q HP 5 mL; Cytiva Life Sciences) with a linear gradient elution from 50–1,000 mM NaCl. Fractions containing processed fusion proteins were then passed over 5 mL of cOmplete His-Tag purification beads to remove residual uncleaved His$_6$-Sumo-PQBP1 and His$_6$-Sumo cleaved tag. The sample was then placed in 6–8 k MWCO dialysis bag while dialyzing against 2 x 2 L of 25 mM Tris pH (8.0 at 23°C), 50 mM NaCl, 1 mM TCEP. The dialysate was purified by Heparin Sepharose chromatography (HiTrap Heparin HP 5 mL; Cytiva Life Sciences) with a linear gradient elution from 50–1,000 mM NaCl. PQBP1 rich fractions were concentrated and purified by Superdex 75 gel filtration chromatography (120 mL;16/600; Cytiva Life Sciences) in 50 mM Tris (pH 8.0 at 23°C) 150 mM NaCl, 1 mM TCEP. Highly pure PQBP1 fractions were pooled, concentrated to ~200 μM, aliquoted and stored at −80°C.

## Endogenous reverse transcription (ERT)

Experimental conditions for ERT reactions were optimized from existing protocols [3,36]. Briefly, we diluted HIV-1$_{SG3}$ΔEnv virions to 50 nM CA in reaction mixtures that contained

50 mM Tris (pH 7.5 at 37°C), 100 mM NaCl, 1.65 mM MgCl$_2$, 40 µM IP6, 1mg/mL bovine serum albumin (BSA), 10 µg/mL melittin (Sigma), ribonucleoside triphosphates (NTPs) (Promega), and deoxynucleotide triphosphates (dNTPs) (Promega). NTP concentrations were chosen to match levels reported in macrophage cytoplasm [94] and were as follows: ATP: 1.124 mM; GTP: 323 µM; UTP: 173 µM; CTP: 25 µM. The concentration of Mg2+ was chosen to match the total concentration of NTPs in the reaction (1.65 mM) and we validated empirically that this optimized ERT yield. dNTP concentrations were previously established [3,95] and were as follows: dATP: 5.2 µM; dGTP: 4.6 µM; dCTP: 6 µM; dTTP: 8 µM. We determined that ERT reactions could proceed using a range of melittin concentrations, and we chose 10 µg/mL for our standard conditions since higher concentrations of melittin (>25 µg/mL) were found to inhibit cGAS activity in the second stage of the assay. Reducing the concentration of melittin below 2.5 µg/mL led to reduced IP6 dependence (i.e., late RT products were produced even in the absence of IP6), suggesting that at lower concentrations of melittin, the membrane was incompletely stripped from the viral core and capsid stability was no longer influenced by IP6.

For plasmid comparisons in ERT reactions, we used the pSG3Δenv plasmid diluted to match the expected copy number for late reverse transcription products from HIV-1$_{SG3}$ (typically 20–40 ng/reaction). Plasmid and HIV-1$_{SG3}$ virions were incubated under identical conditions as described in the figure legends. After 16 h ERT incubations, column-based purification, and qPCR determination of DNA copy number from eluates (described below), plasmid copy numbers ranged from 5 M to 10 M copies per qPCR reaction. ERT reactions were performed in 60 µL volumes and held at 37°C in a ProFlex thermocycler (Thermo Fisher) for the times indicated in the figure legends and cooled to room temperature prior to downstream studies. Standard ERT reactions were performed for 16 h at 37°C. Samples from the same ERT reactions were divided into cGAS sensing assays or used for qPCR analysis of DNA yield.

To compare plasmid control DNA in circular and linear forms, a restriction digest of the pSG3Δenv plasmid was performed (± EcoRI-HF) using rCutSmart buffer (New England Biolabs). Samples were digested for 20 min at 37°C. Undigested and digested plasmid samples were separated on a 0.7% agarose gel for 1 h at 120V before extracting bands corresponding to circular/supercoiled and linearized DNA using a QIAquick Gel Extraction Kit (Qiagen). Circular and linear products were then added to ERT reactions as outlined above.

## Cell-free cGAS sensing assay

Samples from ERT reactions were diluted 50- to 100-fold into cell-free cGAS assay buffers that were similar to standard ERT reaction mixtures, but lacked melittin (50 mM Tris pH 7.5, 100 mM NaCl, 40 µM IP6, 1mg/mL BSA, NTPs, and 2.2 mM MgCl$_2$). Full length recombinant human cGAS was added at 10 nM. cGAS assays were performed for 8 h at 37°C in a thermocycler, with reaction volumes typically ranging from 100–200 µL. We confirmed a linear response in signal for shorter and longer incubations with cGAS (4, 8, and 16 h). BSA was not required for DNA product formation in ERT reactions but was critical to observe a cGAMP signal above the limit of detection in cell-free cGAS assays. Product formation in the cGAS assay was exquisitely sensitive to Mg$^{2+}$ concentration (S1E Fig). Although we observed similar responses to HIV-1 + LEN over a range of Mg$^{2+}$ concentrations (1.65 mM to 5 mM), our standard reaction conditions reported throughout the manuscript were performed using 2.2 mM MgCl$_2$ to ensure adequate signal to noise.

In some experiments, plasmid DNA or HIV-1 ERT samples were heat treated prior to incubation in cell-free cGAS assays. For heat treatment experiments, aliquots from ERT reactions were transferred to new PCR tubes and incubated for 20 s in a ProFlex thermocycler

at a range of temperatures (37°C–62.5°C). Samples were cooled to room temperature before diluting into cGAS assays. For the time course experiments in Fig 2A, all samples were treated with EFV (1 µM) after ERT to prevent additional reverse transcription that may occur during the cGAS assay at 37°C.

After the cGAS assay, samples were held at 4°C for short term storage or frozen at −80°C. cGAMP levels from cGAS assays were determined by ELISA (Arbor Assays) according to the manufacturers' instructions. Reaction mixtures were diluted 1:2 in assay buffer and compared to a standard curve using purified 2'3'-cGAMP. Data were analyzed and plotted using Graph-Pad Prism software. For dose curves comparing responses to LEN, GS-CA1, or PF-74, heat maps were generated using Morpheus (Broad Institute).

## Quantitative PCR analyses

For each ERT reaction, 30 µL of sample was purified using a Qiaquick PCR purification kit (Qiagen) and eluted from the column in 75 µL elution buffer (Qiagen). qPCR reactions were performed and analyzed in triplicate. Primer sets were used to measure early (minus strand strong stop), intermediate (first strand transfer), and late RT (second strand transfer). Standard curves for ERT products were generated by using a 10-fold dilution series of the pSG3Δenv plasmid that were run in parallel with experimental samples and tested using early, intermediate, and late RT primers. Each qPCR sample was comprised of 2.5 µL elution product from purified ERT reactions, 2X SYBR Green Fast qPCR Mix (Abclonal), 1 µL of primers (from a 2 µM working stock), and water to a total of 10 µL reaction volume. qPCR reactions were performed on a QuantStudio 3 (Thermo Fisher) according to the following cycling conditions: 95°C for 10 min, then 45 cycles of 95°C for 10 sec, 55°C for 10 sec, and 72°C for 5 sec, followed by 95°C for 5 sec. A melting curve analysis was then performed going from 65°C to 97°C in 0.11 intervals every 5 sec. Results were analyzed using GraphPad Prism software. For experiments measuring plasmid DNA as a control, plasmid copy numbers determined for each primer set were multiplied by a factor of 1.5 to account for the length difference between the plasmid (14,788 bp) and the reverse transcribed viral genome (~9,730 bp) in order to plot base-adjusted copy number. At least twice as many copies of early and intermediate products are expected for every one copy of late RT in both the plasmid and HIV-1 full-length genome, based on the number of primer binding sites present in those target sequences.

For quantitation of viral cDNA, 100,000 THP-1 cells were seeded in 96 well plates and infected with HIV-GFP (12.5 nM p24) and polybrene (5 ug/mL) in the absence or presence of EFV (10 µM) and LEN. Infected THP-1s were washed 1X with PBS and DNA was extracted using a DNeasy Blood & Tissue Kit (Qiagen). Reverse transcription products were quantified using primers for early, intermediate (unique to HIV-1-GFP), and late RT, and standard curves were generated using a 10-fold dilution series of the HIV-1-GFPΔenv plasmid run in parallel with experimental samples. Each qPCR sample was comprised of 2.5 µL elution product from viral cDNA extraction, 2X SYBR Green Fast qPCR Mix (Abclonal), 1 µL of primers (from a 2 µM working stock), and water to a total of 10 µL reaction volume. qPCR reactions were performed on a QuantStudio 3 (Thermo Fisher) according to the cycling conditions above and data were analyzed using GraphPad Prism software.

For quantification of host gene expression, 100,000 THP-1 cells were seeded in 96 well plates and infected with HIV-1-GFP (12.5 nM p24) plus polybrene (5 µg/mL) in the absence or presence of 10 nM LEN. Infected THP-1 cells were collected in 1.5 mL tubes at 0 h, 16 h, 24 h, 40 h, and 48 h and washed with 1X PBS. Cells were lysed in RLT Buffer (Qiagen), and RNA was isolated using a RNeasy 96 Kit (Qiagen). Normalized quantities of RNA (50 ng) were converted into cDNA using Superscript III (Thermo Fisher). qPCR reactions were carried out

using TaqMan primer probes (Thermo Fisher) and TaqMan Fast Universal PCR Master Mix (Thermo Fisher) on a QuantStudio 3 (Thermo Fisher) in a volume of 10 μL according to the following cycling conditions: 50°C for 2 min, 95°C for 10 min, followed by 40 cycles of 95°C for 15 sec, and 60°C for 1 min. Data were plotted as expression relative to *GAPDH* x 1,000.

### Flow cytometry

Infections with HIV-1-GFP reporter viruses were performed in 96 well plates as described previously [19,89]. Briefly, ~50,000 MDDCs, 25,000 MDMs, or 100,000 THP-1 cells were seeded in 96 well plates and infected as described in the Figure legends with the indicated amounts of virus (typically 12.5 to 50 nM p24) in the presence of polybrene (0.5–5 μg/mL). For flow cytometry, 48 h after infection, MDDCs, MDMs, or THP-1 cells were washed with phosphate buffered saline (PBS, Corning) and then exposed to LIVE/DEAD violet (Thermo Fisher) in PBS for 15 min at 4°C in the dark. For intracellular staining using anti-human ISG15-PE (R&D Systems) or SIGLEC1-647 (BD Biosciences), cells were then fixed and permeabilized using a cytofix/cytoperm kit (BD Biosciences) and stained according to the manufacturers' instructions. Antibody staining was performed for 30 min at room temperature in the dark. Cells were washed and resuspended in PBS with 1% BSA and then analyzed on an Attune flow cytometer (Thermo Fisher). Data were analyzed using FlowJo software (FlowJo LLC). Heat maps were generated using Morpheus (Broad Institute).

### CRISPR-mediated gene editing

THP-1 monocytic cells were modified by lentiCRISPR vectors as previously described [19]. Briefly, we transduced cells in 6-well cluster plates using 800,000 cells per well in 2 mL media with polybrene (5 μg/mL) and concentrated lentiviral stocks (200 μL per well). Two days after transduction, cells were put under selection with puromycin (1 μg/mL, Invivogen) for 4 days and then puromycin was reduced to half-strength for maintenance. Cells were used for experiments beginning 2 weeks after selection. Disruption of target gene expression was confirmed by immunoblot using antibodies to detect cGAS, STING, IRF3, or PQBP1 compared to an actin loading control (as listed in S1 Table).

### Immunoblotting

One million cells were spun down in 1.5 mL tubes and lysed in 150–200 μL SDS sample buffer (2% SDS, 50 mM Tris, 12.5 mM EDTA, plus Halt protease and phosphatase inhibitors (Thermo Fisher)). Samples were sonicated using a microtip Branson sonifier (10–20, 1 s pulses on low output) and protein concentrations were determined using a reducing agent-compatible BCA assay (Thermo Fisher) according to the manufacturer's instructions. Equal amounts of protein (typically 15–20 μg) were loaded with LDS loading buffer (Life Technologies) and separated on 4–12% gradient poly-acrylamide Bolt Bis-Tris gels (Thermo Fisher). Proteins were transferred to nitrocellulose membranes, blocked using 2.5% BSA (Roche) in Tris buffered saline (TBS) with 0.5% tween. Blots were incubated with primary antibodies, then corresponding fluorescent-conjugated anti-mouse or anti-rabbit secondary antibodies (LI-COR), and then fluorescent images were acquired using an Odyssey Imager (LI-COR). Blot images were analyzed using Image Studio Lite (LI-COR).

## Supporting information

**S1 Table. Reagents used in this study.**
(PDF)

**S1 Fig. Capsid stability can be affected by heat or CA mutations. (A)** Schematic depicting heat treatment of plasmid or ERT samples prior to cell-free cGAS sensing assays. **(B)** Copy number of plasmid DNA or viral DNA from ERT reactions that were incubated under identical "standard" conditions (16 h at 37°C) to be taken through limited heat treatment. **(C)** Cell-free sensing assay (related to panels A&B) comparing the ability of plasmid DNA and HIV-1 ERT samples to activate recombinant cGAS. After ERT, samples were incubated briefly (20 s) at 62.5°C before assaying with cGAS under standard conditions (8 h for 37°C). Heat treatment slightly decreased the ability of plasmid DNA to activate cGAS (possibly through partial denaturation and/or aggregation of the DNA into a less accessible structure). In contrast, heat treatment of ERT products robustly increased cGAS activity. **(D)** Gel electrophoresis image depicting a molecular weight ladder (lane 1), circular control DNA (pSG3Δenv plasmid, lane 2), and linear control DNA (pSG3Δenv plasmid digested with EcoRI, lane 3). **(E)** qPCR measurements of DNA copy numbers from circular and linear control DNA samples from (D) that were incubated under standard ERT conditions (16 h at 37°C) in a 2-fold dilution series. **(F)** Cell-free sensing assay (related to panels D&E) of circular and linear control DNA. Samples were incubated with recombinant cGAS and cGAMP levels were measured by ELISA. **(G)** GFP mean fluorescence intensity measured by flow cytometry in THP-1 monocytic cells that were infected for 48 h with HIV-1-GFP, the capsid stabilized CA mutant E45A, or the destabilized mutant Q63/67A at matched virus doses (12.5, 25, and 50 nM p24). Where indicated, LEN (10 nM) was added at the time of infection. For (C), statistics were calculated using an unpaired t test to compare plasmid or HIV-1 samples: $p < 0.05$: *, $p < 0.0001$: ****. Graphs show mean ± SD from three samples from a representative experiment, selected from two independent experiments. Graphics were generated in Illustrator or created in BioRender (https://biorender.com/t60m463,). cGAS–DNA image was adapted from PDB 6CT9 [92]. (TIF)

**S2 Fig. Cell-free sensing of HIV-1 requires virion permeabilization, magnesium, NTPs, and cGAS. (A)** Schematic illustrating that under standard ERT reaction conditions, viral RNA is converted into DNA inside an intact or largely intact capsid. When key components are omitted from the reaction, the ERT reaction is inhibited at the indicated steps. **(B)** qPCR measurements of DNA copy numbers from ERT reactions that were carried out for 16 h at 37°C under standard conditions (including melittin, $MgCl_2$, IP6, NTPs, dNTPs, and additional buffer components as stated in the Methods) compared to reactions performed in the absence of melittin, $MgCl_2$, IP6, or NTPs. **(C)** Models for cell-free sensing reactions corresponding to ERT conditions depicted in (A). Core disruption with LEN permits cGAS sensing of RT products under standard reactions conditions. Sensing is blocked at the indicated steps when specific reaction components are removed. **(D)** Cell-free cGAS sensing assay performed with ERT samples shown in (B). After ERT, samples were incubated with (or without) recombinant cGAS under the conditions shown, with or without LEN (100 nM). cGAMP levels were determined by ELISA. **(E)** Cell-free sensing assay showing sensitivity to $MgCl_2$ concentration. Standard HIV-1 ERT samples were aliquoted into cGAS reactions with the indicated amounts of $MgCl_2$, and with or without LEN (100 nM). **(F)** Cell-free sensing assay depicting core disruption by other capsid inhibitors. ERT samples that were prepared under standard conditions (16 h at 37°C) were aliquoted into cGAS reactions and incubated in the presence or absence of LEN (100 nM), GS-CA1 (100 nM), or PF74 (10 μM). **(G)** Heat map showing dose-dependent increase in cGAS activity for samples incubated with LEN, GS-CA1, or PF74. Samples were prepared for cGAS assays as in (F) with increasing amounts of the indicated capsid inhibitors. Two replicates shown from one of three independent experiments. Statistics were calculated using a one-way ANOVA with Tukey's multiple comparisons test in (D,F) or an unpaired t test

for each concentration of MgCl$_2$ in (E): p < 0.01: **, p < 0.001: ***, p < 0.0001: ****. Graphs depict mean ± SD from three samples from a representative experiment, selected from three independent experiments.
(TIF)

**S3 Fig. Specific CA mutations confer resistance to LEN. (A)** Mean fluorescence intensity of GFP measured by flow cytometry in THP-1 monocytic cells infected for 48 h with HIV-1-GFP compared to LEN-resistant CA mutants M66I and Q67H/N74D at matched virus doses (12.5, 25, and 50 nM p24). Where indicated, LEN (10 nM) was added at the time of infection. **(B)** Cell-free sensing assay of wt, M66I, Q67H/N74D virions. Samples from ERT reactions shown in Fig 3A that were incubated under standard conditions (16 h of ERT at 37°C) were then briefly heat treated (20 s at 62.5°C) before incubating with recombinant cGAS for 8 h at 37°C. cGAMP levels were measured by ELISA. Statistics were calculated using a one-way ANOVA with Tukey's multiple comparisons test: p < 0.001: ***. Graphs depict mean ± SD from three samples from a representative experiment, selected from 2 (A) or 3 (B) independent experiments.
(TIF)

**S4 Fig. LEN augments innate immune responses across a range of virus doses. (A and B)** Flow cytometry of THP-1 monocytic cells infected for 48 h with a range of HIV-1-GFP doses (12.5, 25, and 50 nM p24) and increasing amounts of LEN. GFP is displayed on the x-axis and either SIGLEC1 (A) or ISG15 (B) are shown on the y-axis. Plots represent one of four infection replicates, which are averaged in the heat maps shown in Fig 4C. **(C–E)** Flow cytometry data from THP-1 cells infected with a range of HIV-1-GFP doses (12.5, 25, and 50 nM p24) for 48 h and increasing amounts of LEN (0, 1, 3, 10, 30, and 100 nM), showing SIGLEC1 gMFI (C), ISG15 gMFI (D), or %GFP+ (E). **(F)** Percent SIGLEC1-positive cells measured by flow cytometry. THP-1 monocytic cells were infected for 48 h with HIV-1-GFP compared to LEN-resistant CA mutants M66I and Q67H/N74D at matched doses (12.5, 25, and 50 nM p24). LEN (10 nM) was added at the time of infection. Graphs show mean ± SD from three-four samples from a representative experiment, selected from three (C-E) or two (F) independent experiments.
(TIF)

**S5 Fig. PQBP1 does not affect cGAS activity *in vitro* and is not essential in cells. (A)** SDS-PAGE with Coomassie staining showing a molecular weight ladder (lane 1) and purified recombinant human PQBP1 protein (lane 2). **(B)** Cell-free sensing assay to determine cGAS activity in the presence or absence of PQBP1. Purified PQBP1 was incubated at increasing concentrations with recombinant cGAS alone, or together with HIV-1 samples aliquoted from standard ERT reactions (16 h at 37°C). Cell-free sensing assay reactions were performed under standard conditions. LEN (100 nM) was included as a positive control for viral capsid disruption. **(C)** Immunoblots of THP-1 lysates after lentiCRISPR editing, depicting KO of cGAS or PQBP1 (two independent gRNAs targeting PQBP1), relative to a non-targeting vector control (LCV2). **(D)** Flow cytometry of lentiCRISPR-modified THP-1 cells infected with HIV-1-GFP for 48 h (50 nM p24), depicting GFP expression on the x-axis and SIGLEC1 on the y-axis. **(E)** Flow cytometry of lentiCRISPR-modified THP-1 cells infected with HIV-1-GFP for 48 h (50 nM p24), depicting GFP expression on the x-axis and ISG15 on the y-axis. **(F and G)** Flow cytometry data from THP-1 cells infected with a range of HIV-1-GFP doses (12.5, 25, and 50 nM p24) for 48 h, showing %GFP+ (F) or %ISG15+ (G). To evaluate whether cGAS or PQBP1 have roles in sensing early steps in the virus life cycle, we also tested HIV-1-GFP (50 nM p24) in the presence of the integration inhibitor elvitegravir (EVG, 1 μM). All graphs

show mean ± SD from three samples from a representative experiment, selected from two independent experiments.
(TIF)

**S1 Movie. Model for LEN-mediated disruption of HIV-1 cores and cGAS-STING activation.** A molecular animation depicting an HIV-1 viral core in the cytoplasm of myeloid cells, with reverse transcription proceeding inside an intact or largely intact capsid. In the presence of lenacapavir, accumulated reverse transcripts can become exposed by drug-induced capsid rupturing, enabling viral DNA detection and downstream activation of interferon production via the cGAS-STING-IRF3 pathway.
(MP4)

## Acknowledgments

We thank members of the Johnson and Sundquist laboratories, Owen Pornillos, Barbie Ganser-Pornillos, Jody Puglisi, Elisabetta Viani Puglisi, and other CHEETAH Center investigators for helpful discussions. Oligonucleotides were synthesized by the University of Utah DNA/Peptide Core Facility, sequencing of constructs was performed at the University of Utah DNA sequencing Core Facility, and flow cytometry was performed at the University of Utah Flow Core.

## Author contributions

**Conceptualization:** Wesley I. Sundquist, Jarrod S. Johnson.

**Funding acquisition:** Philip J. Kranzusch, Wesley I. Sundquist, Jarrod S. Johnson.

**Investigation:** Tiana M. Scott, Lydia M. Arnold, Jordan A. Powers, Delaney A. McCann, Ana B. Rowe, Devin E. Christensen, Wen Zhou, Jarrod S. Johnson.

**Methodology:** Tiana M. Scott, Lydia M. Arnold, Devin E. Christensen, Miguel J. Pereira, Wen Zhou, Philip J. Kranzusch, Wesley I. Sundquist, Jarrod S. Johnson.

**Project administration:** Philip J. Kranzusch, Wesley I. Sundquist, Jarrod S. Johnson.

**Supervision:** Philip J. Kranzusch, Wesley I. Sundquist, Jarrod S. Johnson.

**Writing – original draft:** Jarrod S. Johnson.

**Writing – review & editing:** Tiana M. Scott, Lydia M. Arnold, Jordan A. Powers, Delaney A. McCann, Ana B. Rowe, Devin E. Christensen, Miguel J. Pereira, Wen Zhou, Rachel M. Torrez, Janet H. Iwasa, Philip J. Kranzusch, Wesley I. Sundquist, Jarrod S. Johnson.

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
