## [Decision Letter · Decision Letter 0]

7 Jun 2024

Dear Assistant Professor Johnson,

Thank you very much for submitting your manuscript "Cell-free reconstitution of HIV-1 innate immune sensing reveals the capsid is a molecular cloak that protects reverse transcripts from cGAS" for consideration at PLOS Pathogens. As with all papers reviewed by the journal, your manuscript was reviewed by members of the editorial board and by several independent reviewers. In light of the reviews (below this email), we would like to invite the resubmission of a significantly-revised version that takes into account the reviewers' comments.

Dr. Johnson,

Thank you for your patience, reviewers are often a bit late in returning their reviews assigned right around the Cold Spring Harbor Meeting. As you can see, all three reviewers were enthusiastic about the cell free system utilized in your study, which was viewed as innovative and elegant, and I agree. However, the reviewers also noted a number of issues that should be addressed prior to acceptance. Many but not all of these issues are associated with the cellular experiments in your study, with 2 reviewers noting to various degrees the apparent discord between some of your results and previously published studies. While I do not feel that you are obliged to agree with previously published studies in your own study, the reviewers suggest some ways that the timing and dosing of drug or experimental readouts might be modified to allow those in the field to compare other results to the results of your study, and I am inclined to agree some additional experiments in this regard would help accomplish this. Although not noted by reviewers specifically, I also noted that you are achieving a very high level of infection in your THP-1 studies (>80% infection) and would ask that you provide at least one experiment where the cellular responses such as ISG15 or Siglec1 expression are examined at a lower MOI, as I tend to think this is where divergence between study results can occur. In general, I think it is fine for you to use an amount of virus that allows you to monitor the differences you are interested in, so you need not perform every experiment at a lower MOI, but one key result would be useful to those doing research in this area.

We cannot make any decision about publication until we have seen the revised manuscript and your response to the reviewers' comments. Your revised manuscript is also likely to be sent to reviewers for further evaluation.

Sincerely,

Edward M Campbell, PhD

Academic Editor

PLOS Pathogens

Richard Koup

Section Editor

PLOS Pathogens

Michael Malim

Editor-in-Chief

PLOS Pathogens

orcid.org/0000-0002-7699-2064

Dr. Johnson,

Thank you for your patience, reviewers are often a bit late in returning their reviews assigned right around the Cold Spring Harbor Meeting. As you can see, all three reviewers were enthusiastic about the cell free system utilized in your study, which was viewed as innovative and elegant, and I agree. However, the reviewers also noted a number of issues that should be addressed prior to acceptance. Many but not all of these issues are associated with the cellular experiments in your study, with 2 reviewers noting to various degrees the apparent discord between some of your results and previously published studies. While I do not feel that you are obliged to agree with previously published studies in your own study, the reviewers suggest some ways that the timing and dosing of drug or experimental readouts might be modified to allow those in the field to compare other results to the results of your study, and I am inclined to agree some additional experiments in this regard would help accomplish this. Although not noted by reviewers specifically, I also noted that you are achieving a very high level of infection in your THP-1 studies (>80% infection) and would ask that you provide at least one experiment where the cellular responses such as ISG15 or Siglec1 expression are examined at a lower MOI, as I tend to think this is where divergence between study results can occur. In general, I think it is fine for you to use an amount of virus that allows you to monitor the differences you are interested in, so you need not perform every experiment at a lower MOI, but one key result would be useful to those doing research in this area.

Reviewer's Responses to Questions

**Part I - Summary**

Reviewer #1: The study by Scott et al., utilizes a cell free system, that monitors changes in HIV-1 CA stability and how this affects innate immune sensing of HIV-1 RT products through the cGAS-STING pathway. Shielding the viral genome by the capsid shell to evade host sensing is well documented and so the findings in this study are not novel. However, the cell free system is a great tool to understand changes in CA architecture contributing to innate sensing and productive infection. The authors have done a thorough job in explaining most of their findings with illustrations. The novel part of this study is understanding the mode of action of LEN in abolishing infection and eliciting an immune response following treatment. Expect a few additional experiments and clarifications, this is a well-executed work. As mentioned earlier, the concept of HIV_1 capsid as a shield or molecular cloak protecting the genome inside is not that novel and therefore incorporating more data that would address other events in the early life cycle such as RT completion, trigger for capsid disassembly would improve the impact of this work.

Reviewer #2: In this study, Scott et al. report the use of the previously described cell-free reconstituted endogenous HIV-1 reverse transcription system for measuring cGAS sensing of HIV-1 DNA with and without the capsid inhibitor lenacapavir. Interestingly, they show that lenacapavir binding to capsid can promote cGAS activation via the STING pathway in vitro that is similar to results in myeloid cells. This effect was dependent on viral DNA production and was not affected by the presence or absence of PQBP1.

This is a clever system that allows the inclusion of defined factors to interrogate capsid dissociation, reverse transcription, and cGAS sensing. However, the inclusion of several additional measurements, descriptions, and discussion points would improve the manuscript.

Reviewer #3: Scott et al. describe an innovative system to measure the sensing of the HIV-1 genome using purified cores. The system is elegantly developed and highly convincing. Additionally, the investigators demonstrated that LEN treatment on purified viral cores triggers the exposure of the viral DNA, which is then sensed by purified cGAS. However, the data developed in cells is problematic and requires further testing. Investigators should correlate core destabilization during infection of cells by LEN with induction of ISGs.

**Part II – Major Issues: Key Experiments Required for Acceptance**

Reviewer #1: In Fig 1 and Fig S1, the authors bring out the point that CA stability is important to avoid cGAS mediated sensing of viral RT products. In Fig 1G, IP6 at cellular concentrations is sufficient to prevent cGAMP activation in unstable capsids. How does this correspond to viral infection in cells. Is infection restored in unstable capsid at this concentration of IP6?

In lines 90-95, the authors state that CA stability can modulate cGAS sensing, but do not comment on whether host factor recruitment is important to maintain the stability such as IP6, CypA etc.

What is the role of CypA in maintaining CA stability. Can the authors supplement CypA to their assay and see if this avoids cGAS sensing. The results might not be novel but would emphasis on how host factor recruitment is crucial to avoid innate sensing.

Although the authors talk about viral uncoating and viral dsDNA accumulation as a trigger for this process, they do not explain in detail the conclusions from Fig S2E. Increasing the concentration of MgCl2 also elevates cGAMP levels in the absence of LEN. Explanation? Does it increase dsDNA accumulation and possibly rupture? It is still unclear how viral capsid disassembles inside the nucleus and the authors possibly have an experiment in there that could shed some insights into this.

Reviewer #2: 1. What is the definition of "early," "intermediate," and "late" reverse transcripts? The proportion of detected DNA called "late" cDNA (i.e. late relative to early/intermediate) is higher for plasmid compared to HIV-1 in Figure 1B. Why is this? And could this account for lower activation of cGAS and production of cGAMP for virus in Figure 1C and S1C? In Figure 1, it appears that the naked DNA control is purified plasmid, which is presumably mostly circular and supercoiled. Would the kinetics and copy numbers of cGAS activation differ if the plasmid was linearized, which is more similar to the dsDNA viral genome?

2. Some have hypothesized that production of dsDNA during reverse transcription induces capsid uncoating. In Figure 3B, is full-length viral DNA is produced in the majority of the capsids during ERT? If not, could that be the result of a lack of capsid uncoating in this system?

3. Infectivity data in Figure 4 suggests the STING-mediated IFN-I stimulation can occur during infection in the absence of lenacapavir. Is this a result of full completion of reverse transcription? Or a result of other cellular factors? Movie S1 seems to indicate that capsid uncoating does not occur due to completion of reverse transcription (on a microtubule in the cytoplasm?), but this has not been conclusively demonstrated.

4. Lines 163-165: it is stated that high lenacapavir levels "presumably" inhibit reverse transcription in THP-1 cells. This can be easily tested by qRT-PCR. Figure 4C (and S2G): it is unclear why the data are expressed as a heat map instead of a bar graph with error bars like all the other data.

5. It is surprising that the lack of an effect with or without PQBP1 in the Results was not discussed.

Reviewer #3: 1) Measuring interferon at 48 hours is too late. The investigators should perform a time course using different concentrations of LEN 1-100nM. Investigators should consider that LEN at 10 nM potently inhibit infection. Meaning 10nM or less is the relevant concentration that needs to be considered for interpretation of experiments in cells.

2) It is well-documented by various groups, not cited here, that LEN at 10 nM stabilizes the core; therefore, it should not induce interferon; an stabilized core will not expose the viral genome to sensors. The HIV-1 core is stabilized at 5-10 nM during infection (sufficient amount of drug to potently block HIV-1 infection). Destabilization only occurs at concentrations between 30nM-100nM, which is an excessive amount of the drug(overkill). Thus, this data is not in agreement with previous findings. The literature should be considered, as two separate papers using the fate of the capsid assay showed stabilization of the core by LEN at 10 nM.

3) Investigators should measure core stability during infection in the presence of LEN, and correlate this with induction of ISGs in the particular cells that they are using. These experiments will provide information on whether ISGs are induced upon core destabilization during infection.

4) These experiments were likely performed before the FDA approval of LEN. Therefore, it is crucial to repeat the LEN experiments in cells at all concentrations, conducting a time course to measure ISGs.

5) Figure 4B and C: Measuring the induction of interferon in cells at 48 hours is too late. The investigators should perform a time course using different concentrations of LEN (0-48 hours).

6) Figure 4B and C: LEN has passed numerous tests, and no reports have demonstrated the induction of ISGs at 10 nM.

7) Figure 4B and C: this figure needs a control using type I IFN in THP-1 cells and measuring the induction of SIGLEC-1.

Additional Comments:

The manuscript is poorly referenced. Many LEN papers are not cited.

There is a poor description of experimental details in all legends.

Line 28: Many references demonstrating RT in the nucleus are missing.

Figure 1D legend: Include ERT sample preparation details and clarify the 16-hour duration.

Figure 1E legend: Explain when the change of IP6 concentration occurs. Is it after lysis with

Melittin or after the cores have been stabilized in IP6?

Revise all legends for accurate descriptions of experiments, including drug concentrations and experiment timelines.

Line 100: PF74 at the µM range is quite potent against HIV-1, so correct this statement.

The legend for Figure 4G-H is missing experimental details such as time and drug concentration.

**Part III – Minor Issues: Editorial and Data Presentation Modifications**

Reviewer #1: The title for this work mentions an already existing and agreed idea. Can this be reworded?

Reviewer #2: 1. Lines 67, 315, and 494: pHIVSG3 delta env is not "full-length" if it is lacking the env gene.

2. As timing appears to matter according to Figure 3, how long were the ERT reactions and cGAS sensing assays conducted for Figures 1 and 2B?

3. Figure S4: it is unclear why elvitegravir was included in the assay.

Reviewer #3: N/A

PLOS authors have the option to publish the peer review history of their article (what does this mean? ). If published, this will include your full peer review and any attached files.

**Do you want your identity to be public for this peer review?** For information about this choice, including consent withdrawal, please see our Privacy Policy .

Reviewer #1: No

Reviewer #2: No

Reviewer #3: No
---

## [Decision Letter · Decision Letter 1]

27 Dec 2024

PPATHOGENS-D-24-00832R1

Cell-free assays reveal that the HIV-1 capsid protects reverse transcripts from cGAS

PLOS Pathogens

Dear Dr. Johnson,

Thank you for submitting your manuscript to PLOS Pathogens. After careful consideration, we feel that it has merit but does not fully meet PLOS Pathogens's publication criteria as it currently stands. Therefore, we invite you to submit a revised version of the manuscript that addresses the points raised during the review process.

Please submit your revised manuscript within 30 days Feb 25 2025 11:59PM. If you will need more time than this to complete your revisions, please reply to this message or contact the journal office at plospathogens@plos.org. Please include the following items when submitting your revised manuscript:

We look forward to receiving your revised manuscript.

Kind regards,

Edward M Campbell, PhD

Academic Editor

PLOS Pathogens

Richard Koup

Section Editor

PLOS Pathogens

Sumita Bhaduri-McIntosh

Editor-in-Chief

PLOS Pathogens

orcid.org/0000-0003-2946-9497

Michael Malim

Editor-in-Chief

PLOS Pathogens

orcid.org/0000-0002-7699-2064

**Reviewers' Comments:**

Reviewer's Responses to Questions

**Part I - Summary**

Reviewer #1: The authors have satisfactorily addressed the previous review comments. No additional experiments required.

Reviewer #2: My previous comments have been addressed.

**Part II – Major Issues: Key Experiments Required for Acceptance**

Reviewer #1: None

Reviewer #2: None

**Part III – Minor Issues: Editorial and Data Presentation Modifications**

Reviewer #1: None

Reviewer #2: The new title seems incomplete. I would suggest a minor modification: “Cell-free assays reveal that the HIV-1 capsid protects reverse transcripts from cGAS sensing.”

Lines 259-260: “CA-binding proteins that play different roles during entry” is not accurate. I would suggest either revising to “nuclear entry” or “early post-entry replication steps”.

PLOS authors have the option to publish the peer review history of their article (what does this mean? ). If published, this will include your full peer review and any attached files.

**Do you want your identity to be public for this peer review?** For information about this choice, including consent withdrawal, please see our Privacy Policy .

Reviewer #1: No

Reviewer #2: No

**Figure resubmission:**
---

## [Editor Report · Decision Letter 2]

8 Jan 2025

Dear Assistant Professor Johnson,

We are pleased to inform you that your manuscript 'Cell-free assays reveal that the HIV-1 capsid protects reverse transcripts from cGAS immune sensing' has been provisionally accepted for publication in PLOS Pathogens.

Best regards,

Edward M Campbell, PhD

Academic Editor

PLOS Pathogens

Richard Koup

Section Editor

PLOS Pathogens

Sumita Bhaduri-McIntosh

Editor-in-Chief

PLOS Pathogens

orcid.org/0000-0003-2946-9497

Michael Malim

Editor-in-Chief

PLOS Pathogens

orcid.org/0000-0002-7699-2064
---

## [Editor Report · Acceptance letter]

Dear Assistant Professor Johnson,

We are delighted to inform you that your manuscript, "Cell-free assays reveal that the HIV-1 capsid protects reverse transcripts from cGAS immune sensing," has been formally accepted for publication in PLOS Pathogens.

Best regards,

Sumita Bhaduri-McIntosh

Editor-in-Chief

PLOS Pathogens

orcid.org/0000-0003-2946-9497

Michael Malim

Editor-in-Chief

PLOS Pathogens

orcid.org/0000-0002-7699-2064